# A procedure for risk assessment of check dam systems: A case study of Wangmaogou watershed

Lin Wang [1]*, Qiang Zu[1], Qiang Zhang[2]

**1** State Key Laboratory of Eco-hydraulics in Northwest Arid Region of China, Xi'an University of Technology, Xi'an, China, **2** Research Institute of Geotechnical Engineering, China Institute of Water Resources and Hydropower Research, Beijing, China

* linwang@xaut.edu.cn

**Data Availability Statement:** All relevant data are within the manuscript and its Supporting Information files. the GitHub in .csv format, and its URL is: https://github.com/XAUT-WangLin/Summary-table-of-evaluation-index.

## Abstract

Flood-based hydrodynamic damage to check dam systems on the Loess Plateau of China occurs frequently, and there is a strong desire to carry out risk assessments of such check dam systems. This study proposes a weighting method that combines the analytic hierarchy process, entropy method, and TOPSIS to assess the risk of check dam systems. The combined weight-TOPSIS model avoids weight calculation only considers the influence of subjective or objective preference and the bias of the single weighting method. The proposed method is capable of multi-objective risk ranking. It is applied to the Wangmaogou check dam system located in a small watershed on the Loess Plateau. The result of risk ranking matches the reality of the situation. The gray correlation theory model is utilized to rank the risks in the same research area and compared with the results of the combined weight-TOPSIS model. The combined weight-TOPSIS model is more favorable to risk assessment than the gray correlation theory model. The resolution level and decisive judgment of the combined weight-TOPSIS model are more advantageous. These results are in line with the actual conditions. It proves that the combined weight-TOPSIS model can provide a technical reference for the risk assessment of check dam systems in small watersheds.

## Introduction

The Chinese Loess Plateau is located in the middle and upper reaches of the Yellow River. This area is known for its severe soil erosion and high sediment yield [1]. Check dams are an effective long-term engineering measure created by residents in the northwest to control soil erosion on the Loess Plateau [2]. As of the end of 2020, there were 58,776 check dams on the Plateau [3]. Check dams have provided many ecological benefits with regard to silting up land, reducing debris flows, blocking mud and improving the ecological environment [4–10], as shown in Fig 1.

The basic structure of the check dam body has a similar fundamental design to the earth-rock dam. The height of the dam is up to 30 meters. Most check dams use shaft or culvert drainage, and flood discharge capacity is insufficient [11]. Due to the lack of spillways and

**Funding:** The research work is supported by the National Natural Sciences Foundation of China Fund Project (Grant No.51909214), National Key R&D Program Project (2017YFC1501103) provided us with all the funds for the site investigation of the check dam. Shaanxi Water Conservancy Science and Technology Plan Project (2021slkj-9) participated in the data collection and analysis of this study.

**Competing interests:** The authors have declared that no competing interests exist.

limited flood control standards, there is a large gap between earth-rock dams and check dams [12]. Many check dams are full of silt or aging in disrepair.

In recent years, flood-based hydrodynamic damage caused by frequent rainstorms on the Loess Plateau has often caused check dam to breach. Given the unique material composition and geological conditions, once a flood occurs, it will cause unpredictable disasters to the lives and properties of the local people [13–15]. Continuous heavy rainfall in the Yanhe River Basin in Shaanxi induced damage to 516 check dams in July 2013 [16]. In August 2016, an extremely heavy rainfall event occurred in Dalate Banner, Inner Mongolia, and 19 check dams were broken [17]. In July 2017, the disastrous flood on the Wuding River in Shaanxi destroyed 337 check dams in Suide County (Fig 2) [18].

The check dam system is an effective method to control soil erosion and prevent floods on the Loess Plateau [19–22], but once the check dam breaks, it will cause a huge loss of life and property to local residents [23]. A comprehensive risk assessment should be carried out to realize the risk ranking of the check dam system in small basins, which can reduce to the greatest extent the risk caused by the burst of the check dam [24].

The analytic hierarchy process (AHP) is an index weight calculation method that considers subjective factors. It has been widely used in check dams [25], earth-rock dam safety evaluation [26], flood assessment [27], dam site selection [28], land restoration [29], etc. The entropy method is used to calculate the index weight that considers objective factors. It has many applications in the fields of dam risk assessment [30], land use [31], environmental science [32], and other fields. By combining the analytic hierarchy process and entropy method to calculate the combined weight, studies can avoid the bias of a single weighting method. After calculating the combined weight of the evaluation index, the risk of the dam system still needs to be evaluated.

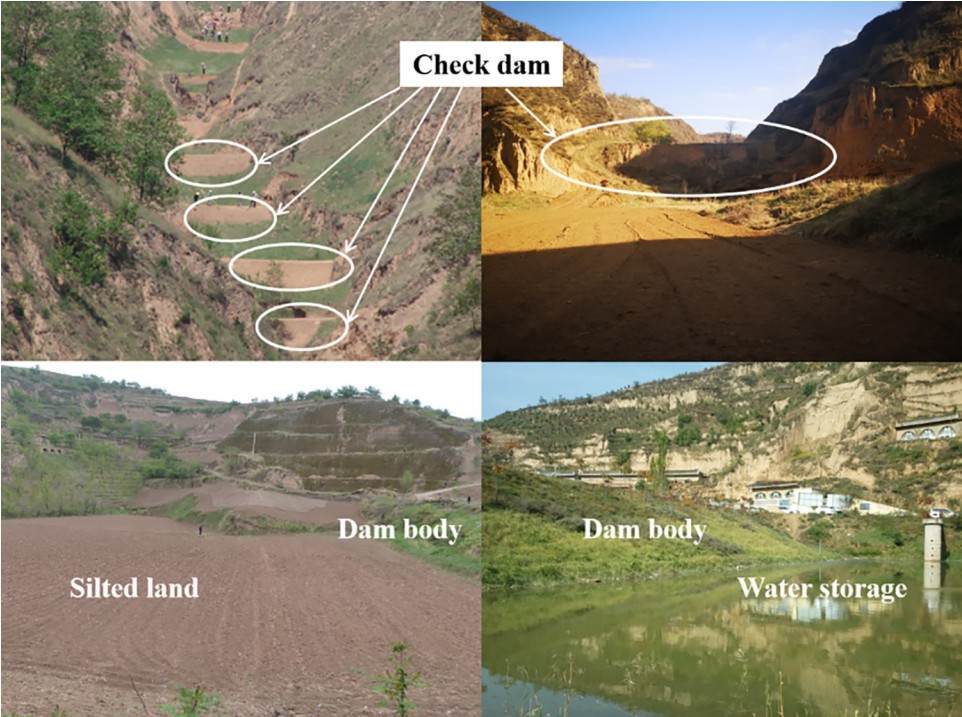

**Fig 1. Check dams with ecological benefits.**

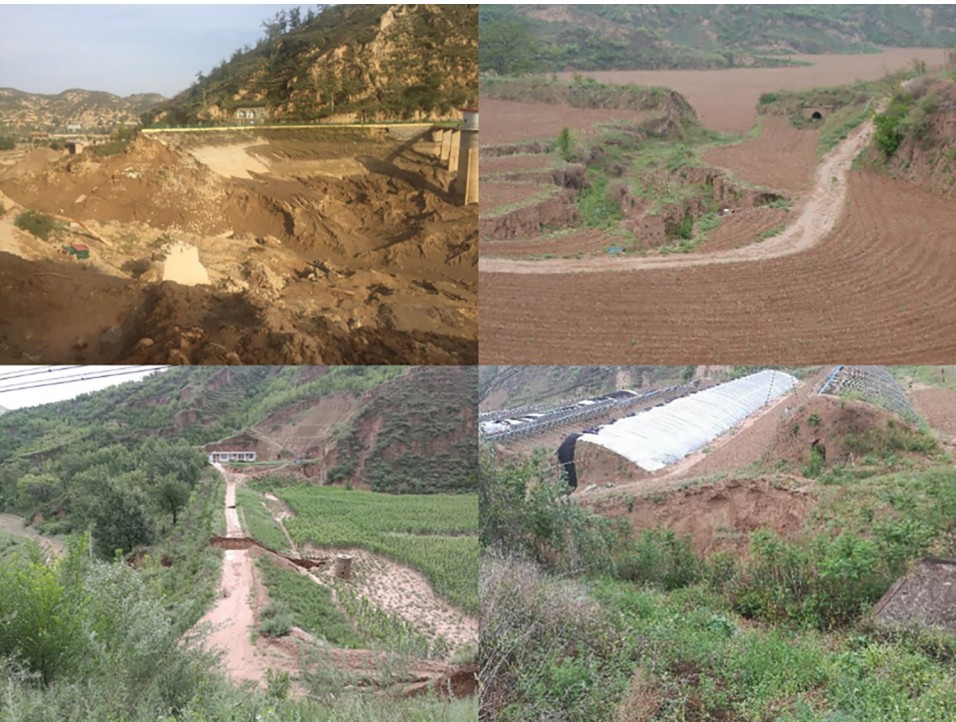

**Fig 2. Breach of check dams in Suide County.**

TOPSIS (a technique for order preference by similarity to the ideal solution), developed by Hwang and Yoon in 1981 [33], is a simple sequencing method in conception and application. This method can analyze principles quickly and perform rapid calculations, and the sample demand is not large. In recent years, it has been used in many fields, such as the evaluation of agricultural water-saving development strategies [34], sustainable building safety analyses [35], and site selection of fixed refuge places in cities and towns [36].

The essence of risk evaluation is to carry out a comprehensive and objective risk ranking under multi-objective situation. This study adopts the subjective and objective combination weight empowerment method combining hierarchical analysis method and entropy weight method was put forward, which aims to avoid the shortcomings of considering only the subjective or the objective factors unilaterally and improve the accuracy of weight determination; The comprehensive risk evaluation model of the check dam system in the small watershed was constructed based on the Technique for Order Preference by Similarity to Ideal Solution (TOPSIS) method, which can realize the risk ranking of check dam system under multiple objectives. It includes three first-level indices of flood, operational and economic risks and ten second-level indices. The combination weight index is established by using the analytic hierarchy process and entropy weight method to obtain the combination weight of the evaluation index, and then combined with the TOPSIS method to carry out the comprehensive risk assessment with the Wangmaogou watershed check dam system as an example. The study is expected to provide a technical reference for the safety prevention and control of check dam systems in small watersheds on the Loess Plateau.

## Theory and methodology

### 1.1 Technical route of the risk assessment model

The analytic hierarchy process and the entropy method are used to establish combined weight indicators with the TOPSIS method [37,38] to carry out risk assessments of check dam systems in small watersheds (Fig 3).

### 1.2 Evaluation index weight

**1.2.1 Analytic hierarchy process.** The analytic hierarchy process is a method of "measurement through pairwise comparisons and relies on the judgments of experts to derive priority scales". Therefore, it is widely used in multiple criteria decision-making tools considering subject factors, such as evaluating the risk of a check dam system in this study [39,40].

The aim of the analytic hierarchy process method is to obtain the weight vector $\omega_{sj}$, which not only considers subject risk evaluation but also satisfies the demand of the consistency check. This subjective weight vector is used to the calculate combined weight $\omega_j$.

The analytic hierarchy process method is specified in 4 steps as follows.

Step 1: Establish a hierarchical structure.

The hierarchical structure is shown in Fig 4 as the risk evaluation index system of the check dam system.

Step 2: Construct the judgment matrix $A_{ij}$ of each level.

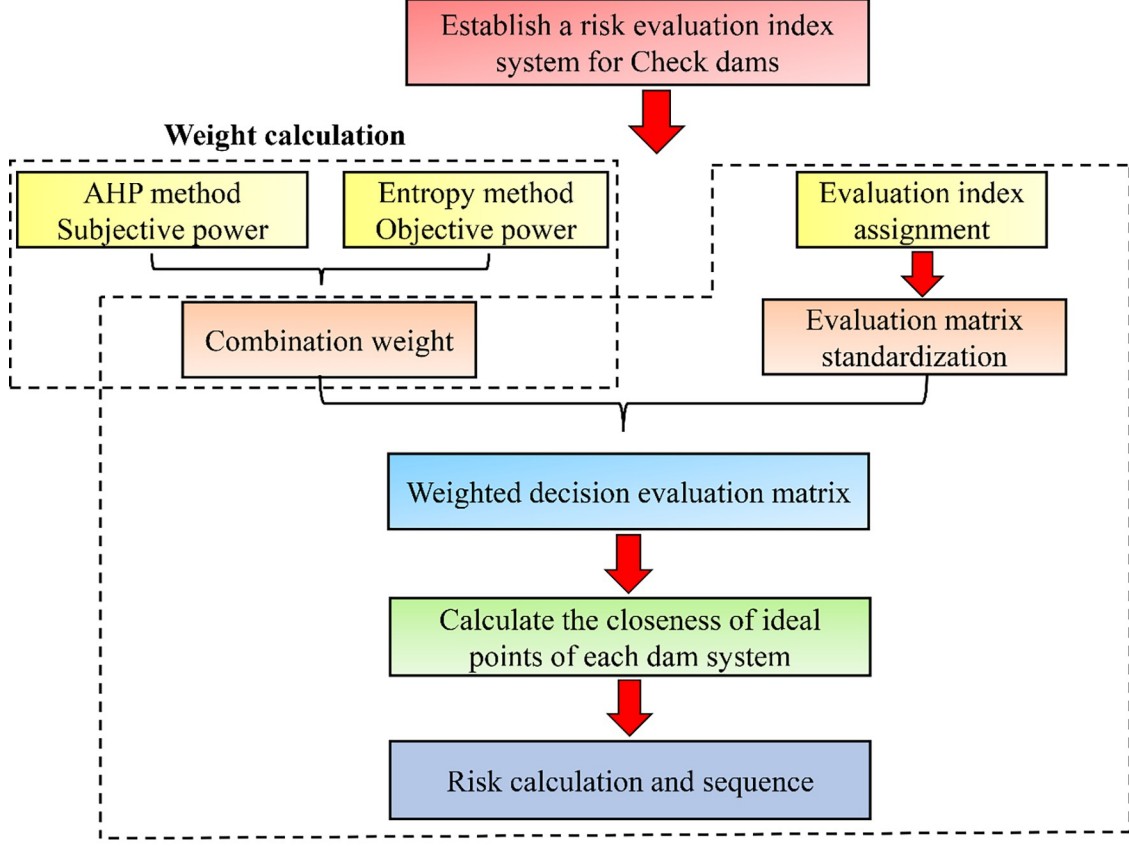

**Fig 3. Technology road map of the combined weight-TOPSIS model.**

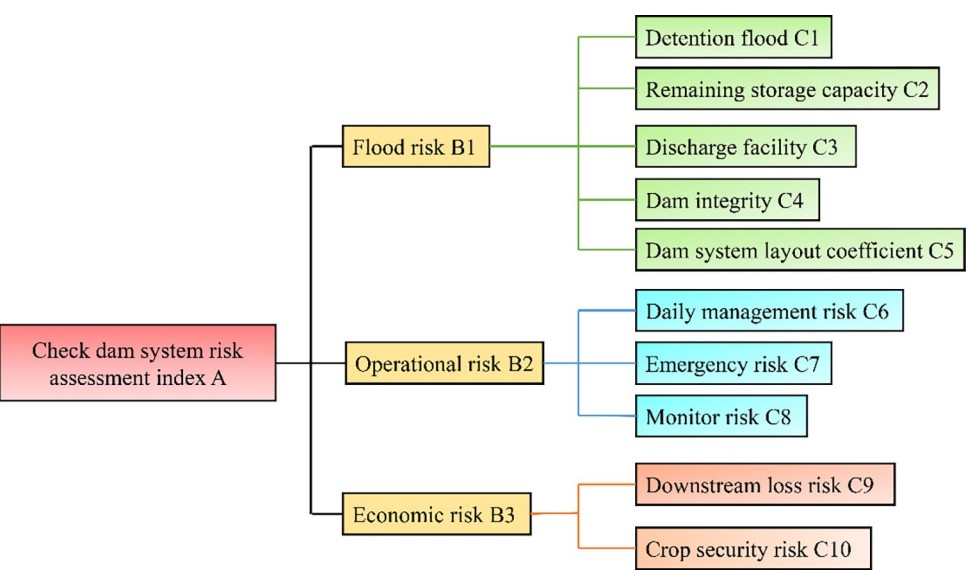

**Fig 4. Risk evaluation index system of check dam systems.**

The judgment matrix is composed of the comparison value of the mutual importance of each evaluation index at the same level.

$$A_{ij} = \begin{bmatrix} a_{11} & a_{12} & \cdots & a_{1n} \\ a_{21} & a_{22} & \cdots & a_{2n} \\ \cdots & \cdots & \cdots & \cdots \\ a_{n1} & a_{n2} & \cdots & a_{mn} \end{bmatrix} \tag{1}$$

where $a_{ij}$ = the scale value, which is expressed by numbers 1–9 and its reciprocal; we give the principle of determining the scale in Table 1.

Step 3: Consistency check.

The consistency index $CI$ is calculated as follows:

$$CI = \frac{\lambda_{\max} - n}{n - 1} \tag{2}$$

**Table 1. Determining principles of scales.**

| Scales | Meaning |
|---|---|
| 1 | Index $i$ is as important as index $j$ |
| 3 | Index $i$ and index $j$ are slightly more important |
| 5 | Index $i$ and index $j$ are obviously important |
| 7 | Index $i$ and index $j$ are strongly important |
| 9 | Index $i$ and index $j$ are extremely important |
| 2, 4, 6, 8 | Indicates the intermediate value of the abovementioned adjacent importance |

Note: When the importance scale of index $i$ over index $j$ is $a_{ij}$, the importance scale of index $j$ over index $i$ is $1/a_{ij}$.

**Table 2. Random consistency index.**

| n | 1 | 2 | 3 | 4 | 5 | 6 | 7 | 8 | 9 | 10 |
|---|---|---|---|---|---|---|---|---|---|----|
| RI | 0 | 0 | 0.58 | 0.90 | 1.12 | 1.24 | 1.32 | 1.41 | 1.45 | 1.49 |

The consistency ratio *CR* is calculated as follows:

where $\lambda_{max}$ = the largest characteristic root of the matrix $A_{ij}$, and $n$ is the order of the judgment matrix $A_{ij}$. The corresponding random consistency index *RI* is determined (Table 2).

$$CR = \frac{CI}{RI} \tag{3}$$

If *CR*<0.1, the consistency of the judgment matrix can be considered acceptable.

Step 4: The eigenvector of the judgment matrix is normalized to be the desired weight vector $\omega_{sj}$.

**1.2.2 Entropy method.**   The entropy method can be used to derive criteria weights objectively from pertinent decision data in case preferential judgments are either partial or unavailable [41]. Using entropy, the weight assigned to a decision criterion is directly related to the average intrinsic information generated by a given set of alternative evaluations at that criterion, as well as to its subjective assessment. Therefore, the entropy method is capable of evaluating the objective weight as a supplement to the analytic hierarchy process method.

The aim of the entropy method is to obtain the weight vector $\omega_{gj}$, which is similar to $\omega_{sj}$ and is also used in calculating of the combined weight $\omega_j$.

The use of the entropy method is specified in 4 steps as follows.

Step 1: Suppose there are a total of $m$ dam systems, each dam system has $n$ evaluation indices, and a judgment matrix $R$ is constructed.

$$R = (r_{ij})_{mn}(i = 1, 2, \ldots, m; j = 1, 2, \ldots, n) \tag{4}$$

where $r_{ij}$ = the value of the jth index of the ith dam system.

Step 2: Normalize the judgment matrix to obtain the normalized judgment matrix $D = (d_{ij})_{mn}$.

Step 3: For the case of $m$ dam systems and $n$ indicators, the entropy $S_j$ of the evaluation indicators can be determined as:

$$S_j = -\frac{\sum_{i=1}^{m} f_{ij}\ln f_{ij}}{\ln m}(i = 1, 2, \ldots, m; j = 1, 2, \ldots, n) \tag{5}$$

where $f_{ij} = (1 + d_{ij})/\sum_{i=1}^{m}(1 + d_{ij})$.

Step 4: The entropy weight vector $W$ of the evaluation index is:

$$W = (\omega_{gj})_{1 \times n} = \left( \frac{1 - S_j}{n - \sum_{j=1}^{n} S_j} \right)_{1 \times n} \tag{6}$$

where $\omega_{gj}$ = the entropy weight of the jth evaluation index.

**1.2.3 Combination weight.** The subjective weighting $\omega_{sj}$ and the objective weighting $\omega_{gj}$ are integrated to obtain the combined weight $\omega_j$:

$$\omega_j = \frac{(\omega_{sj} \cdot \omega_{gj})^{0.5}}{\sum\limits_{j=1}^{m} (\omega_{sj} \cdot \omega_{gj})^{0.5}} \quad (j = 1, 2, \cdots, m) \tag{7}$$

## 1.3 TOPSIS evaluation method

The evaluation steps are divided into the following five steps:

Step 1: Standardization of the indicator data.

For the case of $m$ dam systems and $n$ evaluation indicators, the initial evaluation matrix $X = (x_{ij})_{mn}$ can be obtained by referring to Formula (1). The data in $X$ are standardized.

When a lager positive index is better, the positive index $P_{ij}$ calculation formula is as follows:

$$P_{ij} = \frac{x_{ij} - \min(x_{ij})}{\max(x_{ij}) - \min(x_{ij})} \tag{8}$$

When a smaller negative index is better, the positive index $P_{ij}$ calculation formula is as follows:

$$P_{ij} = \frac{\max(x_{ij}) - x_{ij}}{\max(x_{ij}) - \min(x_{ij})} \tag{9}$$

The final standardized matrix is $P = [p_{ij}]_{m \times n}$.

Step 2: Establish a weighted decision evaluation matrix.

The weighted normalized matrix $V$ after considering the weight of each evaluation index is:

$$V = P \cdot W = [v_{ij}]_{m \times n} \tag{10}$$

where $W$ = the evaluation index weight vector. This study adopts subjective and objective combination weights, as calculated by Formula (7).

Step 3: Determine the positive and negative ideal solutions $V^+$ and $V^-$.

$$V^+ = \{\max(v_{ij}) | i = 1, 2, \ldots, m\} = \{v_1^+, v_2^+, \ldots, v_n^+\} \tag{11}$$

$$V^- = \{\min(v_{ij}) | i = 1, 2, \ldots, m\} = \{v_1^-, v_2^-, \ldots, v_n^-\} \tag{12}$$

Step 4: Calculate the distance.

The Euclidean distances from the evaluation index to the positive and negative ideal solutions $V^+$ and $V^-$ are $D_i^+$ and $D_i^-$.

$$D_i^+ = \sqrt{\sum_{j=1}^{n} (v_j^+ - v_{ij})^2} \, (i = 1, 2, \ldots, m) \tag{13}$$

$$D_i^- = \sqrt{\sum_{j=1}^{n} (v_{ij} - v_j^-)^2} \, (i = 1, 2, \ldots, m) \tag{14}$$

where $v_j^+$ = the positive ideal point of evaluation index $j$ and $v_j^-$ = the negative ideal point of

evaluation index $j$. The closer the evaluation indicator is to the positive ideal point, the better the indicator and the lower the risk will be.

Step 5: Calculation and sorting of relative closeness.

The relative closeness $Ti$ is used to indicate the degree of the evaluation value and the optimal value. The larger the value is, the closer it is to the optimal value, i.e., the lower the risk. The calculation formula is as follows:

$$T_i = \frac{D_i^-}{D_i^+ + D_i^-} (i = 1, 2, \ldots, m) \tag{15}$$

The closeness of each dam system is calculated and sorted to obtain the relative risk of each dam system.

## Risk evaluation index system

### 2.1 Construction of the risk evaluation index system for check dam systems

First, the risk evaluation index system of check dam systems can be established to evaluate the system risk. The establishment of this system needs to identify the risk factors for check dams. Through the investigation of a large number of water-damaged check dams and dam systems, the common causes of water damage to check dams can be determined [42–44].

These factors include the following:

1. Dams that are suffering from excessive rainstorms and floods exhibit greater risk.

2. The effective storage capacity is full, and the flood detention capacity is insufficient.

3. Unmatched drainage engineering facilities lead to poor drainage.

4. The dams are of poor construction quality.

5. There is a lack of control over backbone projects in the watershed, the dam system layout is unreasonable, and chain dam failures are likely to occur.

6. Later management and protection are relatively weak.

Based on the historical water damage, considering that the check dam provides the special function of flood detention and silt retention and silt land reclamation, its water damage will cause submergence losses to downstream facilities and economic crops. The risk assessment of the check dam system is carried out from the three perspectives of external man-made management strategies, protection measures and economic losses. Based on the established system of predecessors [45–47], three first-level indicators that affect the safety of the check dam system (flood disaster, operation, economy) and a risk evaluation index system with 10 secondary indicators are established (Fig 4 & Table 3).

### 2.2 Valuation method of the risk evaluation index for check dam systems

Based on the survey data, 10 indicators in the risk evaluation index system are quantitatively assigned [48]. To eliminate the dimensional influence, the evaluation index matrix is standardized after the assignment. To reasonably reflect the degree of each evaluation index influence on the risk of the dam system, it is necessary to calculate each index weight vector [48]. The weighted evaluation matrix is the decision evaluation matrix. In the flood risk layer, evaluation index C3 is assigned to the water release facility. The value is 1 when the main dam has a complete spillway, 0.9 when the spillway has a small amount of damage, 0.5 when horizontal pipes and vertical wells are used for water discharge, and 0.1 when the discharge port is blocked and

**Table 3. Connotation of risk evaluation indices for check dam systems.**

| First-level index | Secondary indicators | Indicator meaning | Index nature |
|---|---|---|---|
| Flood risk (B1) | Detention flood (C1) | The runoff of the rainstorm within a certain return period within the control area of the check dam minus the amount of discharge from the discharge structure | Negative index |
| | Remaining storage capacity (C2) | The capacity of the check dam to withstand floods, the larger the remaining storage capacity, the stronger the ability to withstand floods | Positive index |
| | Drain facility (C3) | Including the discharge capacity of vertical shafts, horizontal pipes, and spillways | Positive index |
| | Dam integrity (C4) | The extent of damage to the check dam body | Positive index |
| | Dam system layout coefficient (C5) | The rationality of the dam system layout, the more reasonable the layout, the larger the coefficient | Positive index |
| Operational risk (B2) | Daily management (C6) | Whether there is a corresponding department to manage and maintain the check dam | Positive index |
| | Emergency response (C7) | Whether there are emergency measures for accidents | Positive index |
| | Monitoring facility (C8) | Whether there are correspondingly complete and normal monitoring facilities | Positive index |
| Economic risk (B3) | Downstream loss risk (C9) | Whether there are roads, houses, factories, etc. downstream to determine the indicators of downstream economic risks | Negative index |
| | Crop security risk (C10) | Allowable submerged water depth of dam system crops is higher than the depth of stagnant flood | Negative index |

cannot be discharged. An evaluation index of integrity degree C4 is assigned to the dam body, and the value is 1 when the main dam body is intact, 0.8 when there are real cracks, and 0.2 when there are small caves. The evaluation index dam system layout coefficient C5 reflects the rationality of the dam system layout, and the value ranges from 0 to 1. A value of more than 0.65 indicates a reasonable layout, a value less than 0.4 indicates an unreasonable layout, and values in the middle indicate a reasonable layout.

In the operation risk layer, the evaluation index daily management risk C6 is assigned. When the relevant department maintains the check dam, the value is 1; otherwise, it is 0.1. The evaluation index emergency risk C7 is assigned. When there is an accident, the value is 1 for emergency measures; otherwise, it is 0. A value is assigned to the evaluation index monitoring risk C8. When there are complete and normal monitoring facilities, the value is 1; otherwise, it is 0.1.

In the economic risk layer, the evaluation index downstream loss risk of C9 is assigned. When there are important residential buildings downstream, the backbone dam system unit is set to 1, and the branch dam system unit is set to 0.8. The evaluation index crop yield risk C10 is assigned, and its risk value is related to the flood return period corresponding to the stagnant flood depth.

## Risk assessment of check dam systems—a case study of Wangmaogou watershed

### 3.1 Study area

Wangmaogou is located in Suide County, Yulin City, Shaanxi Province. It is a secondary branch of Jiuyuangou located on the left bank of the Wuding River. The geographical position is 940~1188 m east longitude, the drainage area is 5.97 km$^2$, and the main ditch length is 3.75 km. The slope is generally above 20°. The amount of rainfall in the basin is small, and it is unevenly distributed. The average annual rainfall is 513 mm, and the rainfall in the flood

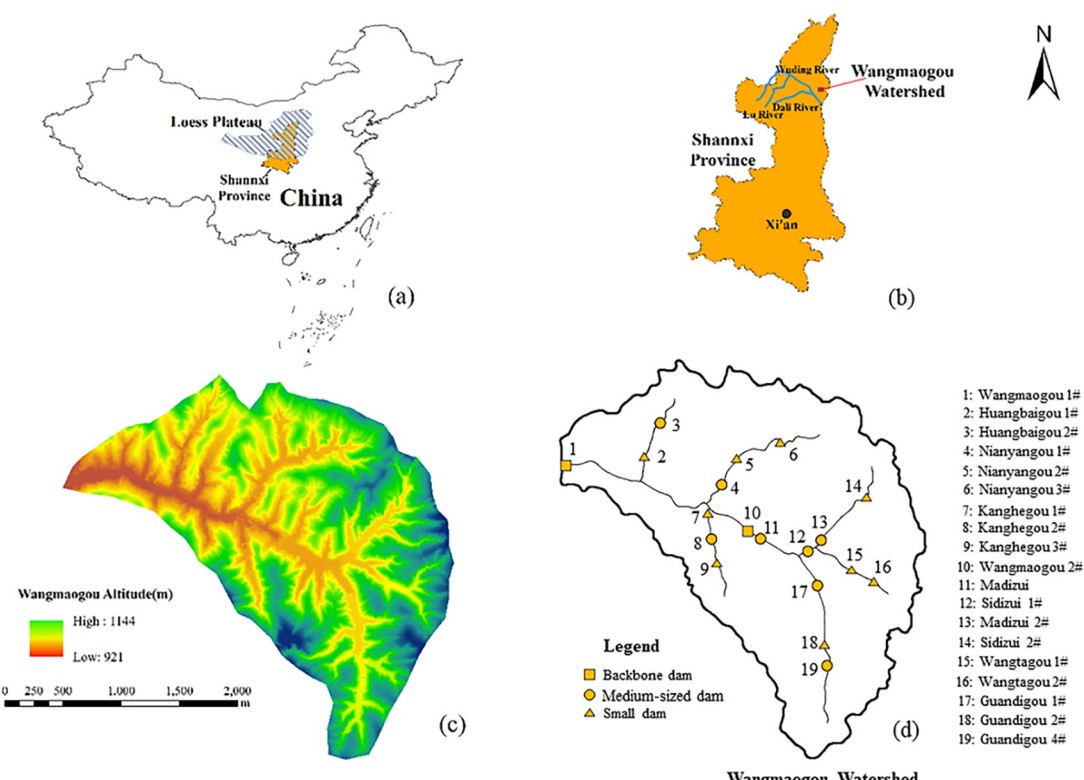

**Fig 5.** Location of the area (a, b); the altitude of the Wangmaogou watershed (c, Drawn by ESRI's ArcGIS); the layout of check dam types (d).

season accounts for more than 70% of the total annual rainfall [49,50]. In 2012 and 2017, flood-based hydrodynamic damage to the check dam system caused by frequent rainstorms occurred; thus, risk analysis is urgently needed. According to the data from the Suide Soil and Water Conservation Scientific Experiment Station of the Yellow River Conservancy Commission, there are 22 check dams in the Wangmaogou watershed, including 2 backbone dams, 8 medium dams, and 12 small dams [51]. The check dam that breached after flood-based hydrodynamic damage in the Wangmaogou watershed on July 15, 2012 was used as an example to carry out the research. After excluding the check dams that were breached and full before 2012, 19 check dams were selected for analysis, as shown in Fig 5. This study divides Wangmaogou check dam systems into the Guandigou unit, Wangtagou unit, Wangmaogou Unit 2, Nianyangou unit, Kanghegou unit and Huangbaigou unit.

## 3.2 Determination of the index weight

The risk evaluation index approach of the check dam system is shown in Fig 5. Based on the 3 first-level and 10 second-level indicators, when the Wangmaogou watershed encountered a once-in-a-hundred-year heavy rainfall event, the data used in the risk evaluation index are shown in Table 4.

Among the evaluation indicators, detention floods, downstream economic risks, and income-guaranteed risks are negative indicators, and the rest are positive indicators. The matrix $P$ is obtained from the initial evaluation matrix standardized, which is based on the assignment results of the dam system layout coefficient, the downstream economic risk, and

**Table 4. Summary table of evaluation index.**

| | Detention flood/$10^4$ m³ | Remaining storage capacity/$10^4$ m³ | Drain facility | Dam integrity | Daily management | Emergency measures | Monitor facility |
|---|---|---|---|---|---|---|---|
| Guandigou Unit | 6.57 | 43.29 | Spillway | Tiny cave | No | No | No |
| Wangtagou Unit | 4.26 | 5.23 | No | intact | No | No | No |
| Wangmaogou Unit 2 | 5.53 | 64.38 | Shaft | intact | Yes | Yes | Yes |
| Nianyangou Unit | 4.93 | 32.76 | Lying tube | intact | No | No | No |
| Kanghegou Unit | 2.28 | 9.67 | Blocked pipe | Rills, cracks | No | No | No |
| Huangbaigou Unit | 1.48 | 3.85 | Shaft | intact | No | No | No |

the income-guaranteed risk:

$$P = \begin{bmatrix}
0.000 & 0.652 & 1.000 & 0.000 & 0.178 & 0.000 & 0.000 & 0.000 & 1.000 & 0.030 \\
0.454 & 0.023 & 0.000 & 1.000 & 0.089 & 0.000 & 0.000 & 0.000 & 1.000 & 0.000 \\
0.204 & 1.000 & 0.444 & 1.000 & 1.000 & 1.000 & 1.000 & 1.000 & 0.000 & 0.020 \\
0.322 & 0.478 & 0.444 & 1.000 & 0.333 & 0.000 & 0.000 & 0.000 & 1.000 & 0.677 \\
0.843 & 0.096 & 0.000 & 0.750 & 0.000 & 0.000 & 0.000 & 0.000 & 1.000 & 0.687 \\
1.000 & 0.000 & 0.444 & 1.000 & 0.267 & 0.000 & 0.000 & 0.000 & 1.000 & 1.000
\end{bmatrix}$$

According to the principle of scale determination, a judgment matrix is constructed which meets the consistency requirement. Then, the eigenvector of the judgment matrix is standardized to obtain the subjective weight vector from the analytic hierarchy process. Third, the entropy weight of each evaluation index, i.e., the objective weight, is calculated by Formula (5) and Formula (6). Finally, the combined weight vector is obtained based on the Formula (7) from the subjective weight vector and the objective weight vector. The evaluation index weight vector is shown in Fig 6, which is calculated by the analytic hierarchy process, entropy weight method, and combined weight method. The combination weight value lies between the subjective and objective weight values, indicating the effect of information integration.

## 3.3 Risk ranking

The standardized evaluation matrix $P$ is weighted to obtain the weighted decision evaluation, and matrix $V$ is obtained as shown below.

$$V = P \cdot W$$
$$= \begin{bmatrix}
0.000 & 0.057 & 0.092 & 0.000 & 0.012 & 0.000 & 0.000 & 0.000 & 0.101 & 0.004 \\
0.044 & 0.002 & 0.000 & 0.058 & 0.006 & 0.000 & 0.000 & 0.000 & 0.101 & 0.000 \\
0.020 & 0.088 & 0.041 & 0.058 & 0.068 & 0.116 & 0.127 & 0.118 & 0.000 & 0.003 \\
0.031 & 0.042 & 0.041 & 0.058 & 0.023 & 0.000 & 0.000 & 0.000 & 0.101 & 0.091 \\
0.082 & 0.008 & 0.000 & 0.043 & 0.000 & 0.000 & 0.000 & 0.000 & 0.101 & 0.092 \\
0.097 & 0.000 & 0.041 & 0.058 & 0.018 & 0.000 & 0.000 & 0.000 & 0.101 & 0.134
\end{bmatrix}$$

where $W$ = the combination weight, and $P$ = *the standardized evaluation matrix*.

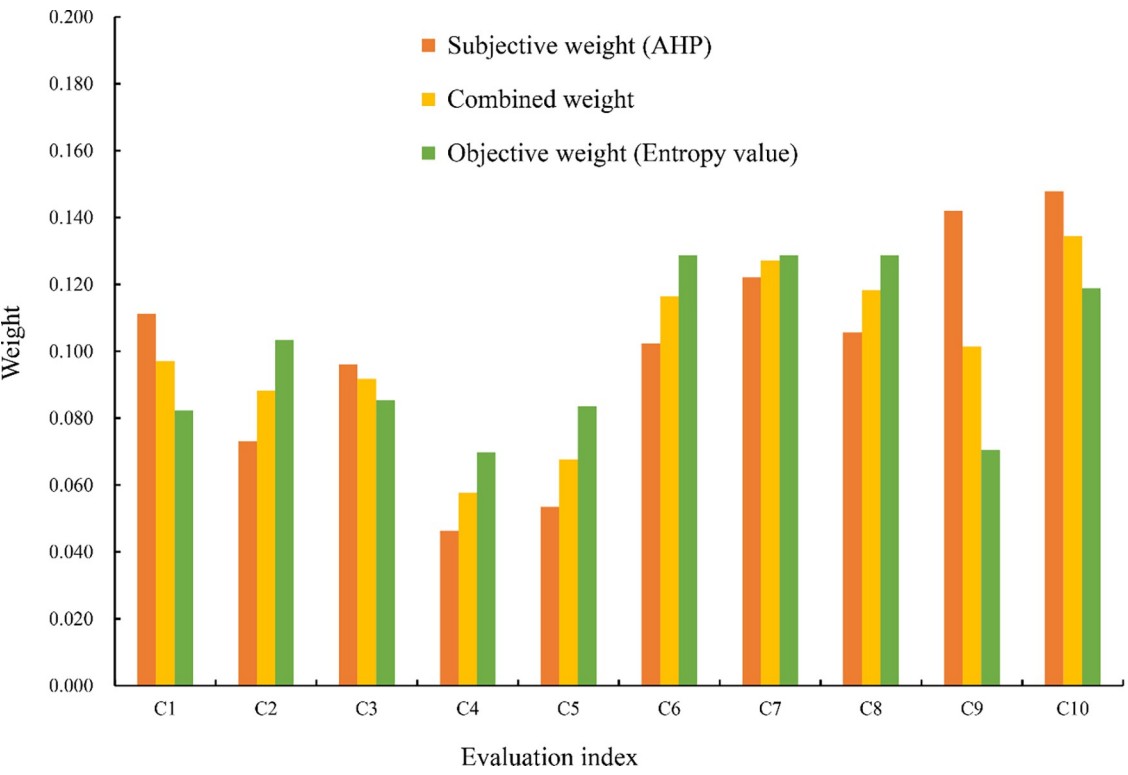

**Fig 6. The weight of the check dam system risk evaluation index in the Wangmaogou watershed.**

The positive and negative ideal solutions $V^+$ and $V^-$ are calculated by Formula (11) and Formula (12):

$$V^+ = \begin{bmatrix} 0.097 & 0.088 & 0.092 & 0.058 & 0.068 & 0.116 & 0.127 & 0.118 & 0.101 & 0.134 \end{bmatrix}$$

$$V^- = \begin{bmatrix} 0 & 0 & 0 & 0 & 0 & 0 & 0 & 0 & 0 & 0 \end{bmatrix}$$

The distances of $D^+$ and $D^-$ from each dam system unit to the positive and negative ideal points are calculated by Formula (13) and Formula (14):

$$D^+ = \begin{bmatrix} 0.278 & 0.290 & 0.190 & 0.238 & 0.255 & 0.238 \end{bmatrix}$$

$$D^- = \begin{bmatrix} 0.149 & 0.125 & 0.248 & 0.164 & 0.166 & 0.208 \end{bmatrix}$$

Based on the combined weight-TOPSIS model to calculate the relative closeness $T$ of each dam system unit, the following can be obtained:

$$T = \begin{bmatrix} 0.349 & 0.301 & 0.566 & 0.408 & 0.393 & 0.466 \end{bmatrix}$$

## Results and analysis

### 4.1 Results and analysis for risk assessment of check dam systems

At present, there is a lack of comprehensive evaluation index systems and general evaluation methods for check dam systems in small watersheds. Due to the limited data, there are no

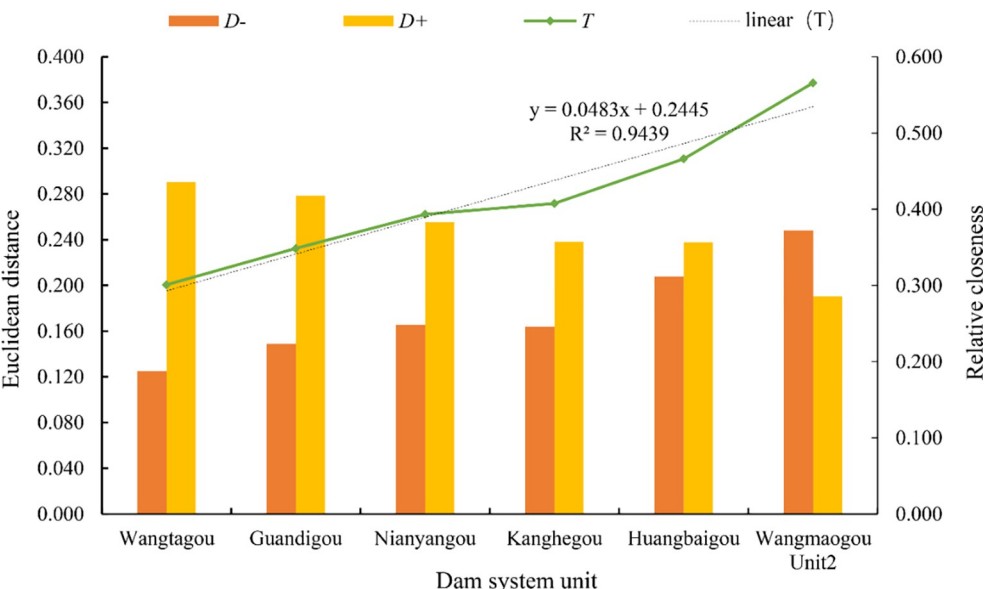

**Fig 7. Euclidean distance and relative closeness degree of check dam system units in the Wangmaogou watershed.**

direct data that can reflect the risk assessment of check dam systems. On-site investigation of the Wangmaogou dam system unit, it can reflect the risk level, and provide a risk assessment of check dam systems.

The evaluation results are summarized as follows.

(1) The higher the relative closeness is, the lower the risk. The risk ranking of Wangmaogou small watershed dam system units is conducted by using the combined-weight-TOPSIS risk assessment model. The result is Wangmaogou Unit 2<Huangbaigou Unit<Nianyangou Unit<KangHegou unit<Guandigou unit<Wangtagou unit (Fig 7). Fitting the relative closeness in Fig 7, the coefficient of determination $R^2$ after fitting is 0.9439, which indicates that the distribution of evaluation values calculated by the combined weight-TOPSIS model is uniform and reasonable.

(2) The Wangtagou unit has the greatest risk. The Wangtagou unit is located in a branch of the Wangmaogou watershed, and the risk is greater than that of the backbone dam Wangmaogou No. 2 Dam. After conducting field surveys in the Wangmaogou small watershed, it is found that the dam heights of the Wangtagou No. 1 and No. 2 dams of the Wangtagou unit dam system are 9 m and 13 m, respectively. The widths are 3.6 m and 4 m, without any drainage facilities, and the drainage is not smooth. The distances between the bottom and the top are 0.1 m and 4.7 m, respectively. The remaining storage capacity is relatively small, and the risk is higher. Therefore, it is necessary to improve the drainage structure construction of branch check dams.

(3) The construction standard of Unit 2 in Wangmaogou, i.e., the backbone dam of the river basin, is the highest, and its risk is the lowest. According to the measured data from the Suide Water Conservation Station, during flood-based hydrodynamic damage caused by the heavy rainfall that hit the Wangmaogou watershed on July 15 in 2012, a total of 8 dams breached, and the Wangtagou No. 1 dam and No. 2 dam in the Wangtagou unit dam

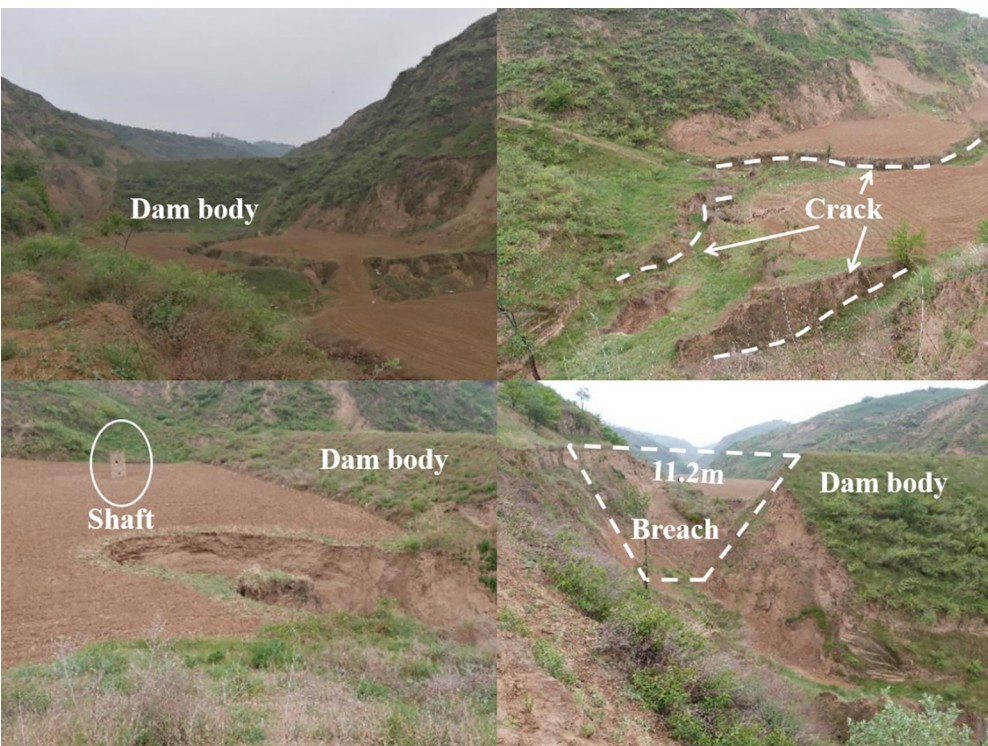

**Fig 8. The Breach of Wangmaogou No. 2 backbone dam.**

system breached the width of Wangtagou No. 1 dam breach was 1 m, and the width of dam No. 2 was 11.2 m (Fig 8). The Wangmaogou No. 2 dam did not breach. The risk of the No. 2 dam is lower than that of the Wangtagou unit.

According to the actual survey after the "7.15" rainstorm [51]. Neither of the two check dams in Unit 2 of Wangmaogou broke during the rainstorm. In the Huangbaigou Unit, the Huangbaigou #2 dam did not break, and the Huangbaigou #2 dam break is circular with a breach of 2m. There are three check dams in the Nianyangou Unit, and two of them have broken. There are also three check dams in the Kanghegou Unit, and all of them have broken. Among them, only the #3 dam of Kanghegou has broken up to 7 m wide, and the breaking is relatively serious. The two check dams in Wangtagou Unit broke seriously, and the maximum breach reached 11.2 m wide." Through the actual investigation after the rainstorm, the evaluation result "Wangmaogou Unit 2<Huangbaigou Unit<Nianyangou Unit<KangHegou unit<Guandigou unit<Wangtagou unit" of the siltation dam unit is verified to be correct.

Above all, the risk ranking results are consistent with the actual situation.

## 4.2 Comparison with existing risk assessment models

(1) The entropy, AHP and combined weight-TOPSIS model, are used to carry out a comparative analysis of check dam systems risk assessment in small watersheds. The ideal points' relative closeness and the risk ranking results of each dam system unit are shown in Table 5. The Spearman rank correlation coefficient method is used to test the correlation degree [52]. The correlation coefficient between the combined weight-TOPSIS model and

**Table 5. Closeness degree and risk ranking result.**

| Dam system unit name | Combination weight-TOPSIS | | AHP-TOPSIS | | Entropy Method-TOPSIS | |
|---|---|---|---|---|---|---|
| | Closeness | Risk ranking | Closeness | Risk ranking | Closeness | Risk ranking |
| Guandigou Unit | 0.349 | 2 | 0.396 | 2 | 0.315 | 2 |
| Wangtagou Unit | 0.301 | 1 | 0.360 | 1 | 0.262 | 1 |
| Wangmaogou Unit 2 | 0.566 | 6 | 0.493 | 5 | 0.632 | 6 |
| Nianyangou Unit | 0.408 | 4 | 0.462 | 4 | 0.337 | 3 |
| Kanghegou Unit | 0.393 | 3 | 0.461 | 3 | 0.370 | 4 |
| Huangbaigou Unit | 0.466 | 5 | 0.530 | 6 | 0.412 | 5 |

the subjective weight-TOPSIS model ranking result is 0.943. The correlation coefficient between the combined weight-TOPSIS model and the objective weight-TOPSIS model ranking result is 1.000. The correlation is extremely high, and the evaluation results are highly consistent, indicating that the combined weight-TOPSIS model achieves subjective and objective data information. Considering the subjective and objective weights, the correlation coefficient of the TOPSIS model ranking results is 0.943, indicating the difference between subjective judgment and objective information.

(2) The combined weight-TOPSIS model and gray relational theory [53] are usually used to carry out a comparative evaluation of dam systemic risks in the same study area. The results are shown in Table 6. Both methods are simple to operate and require quick calculations. The ranking results obtained by the combined weight-TOPSIS model and the gray relational analysis are consistent, indicating that the risk evaluation model proposed is reasonable and reliable. A linear fitting is performed on the ranking value of the posting progress, and it is considered that the slope value can explain the overall resolution level of the ranking result [54]. After calculation, the linear fitting functions of the combined weight-TOPSIS model and the gray relational analysis model's post schedule value are $y = 0.0483x + 0.2445$ and $y = 0.0263x + 0.0925$ (Fig 9), respectively. The slope values are 0.0483 and 0.0263, respectively. Therefore, the combined weight-TOPSIS model is better at the evaluation and resolution levels and is more conducive to decision-making and judgment.

(3) Subjective weight and objective weight contribute equally to the combined weight. The weight can be further optimized by adjusting the degree of contribution of the subjective weight and objective weight. After subsequent calculations, it is found that when different subjective weights and objective weights are used, such as 0.3: 0.7, 0.4: 0.6, 0.5: 0.5, 0.6: 0.4, and 0.7: 0.3, the risk ranking results are consistent. After a linear fitting is made on the

**Table 6. Comparison of closeness degree and risk ranking results between the combination weight-TOPSIS and gray relational analysis.**

| Dam system unit name | Combination weight-TOPSIS | | Gray relational analysis | |
|---|---|---|---|---|
| | Closeness | Risk ranking | Evaluation value | Risk ranking |
| Guandigou Unit | 0.349 | 2 | 0.154 | 2 |
| Wangtagou Unit | 0.301 | 1 | 0.136 | 1 |
| Wangmaogou Unit 2 | 0.566 | 6 | 0.303 | 6 |
| Nianyangou Unit | 0.408 | 4 | 0.170 | 4 |
| Kanghegou Unit | 0.393 | 3 | 0.158 | 3 |
| Huangbaigou Unit | 0.466 | 5 | 0.187 | 5 |

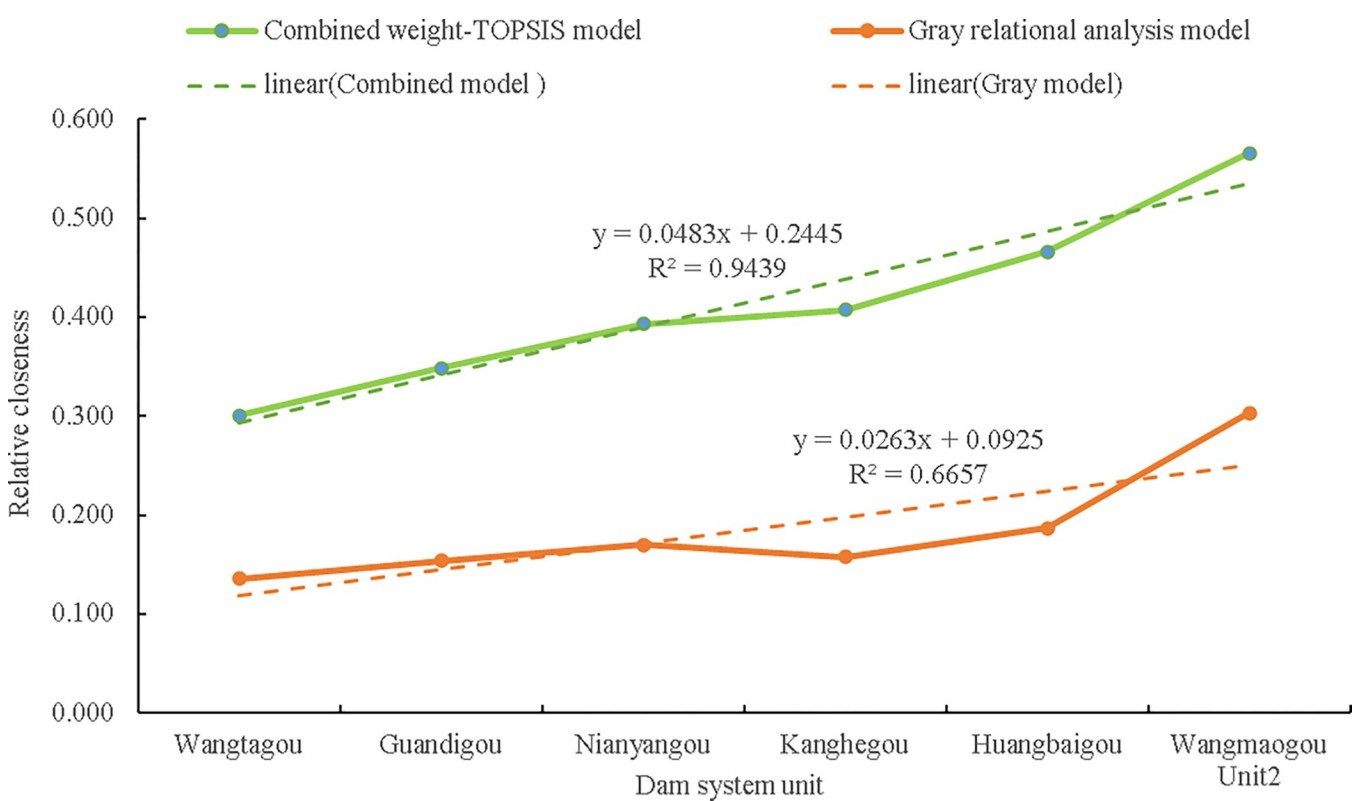

**Fig 9. Comparison of fitting results of correlation closeness between the combined weight-TOPSIS model and gray theory model.**

sorted ideal point-posting progress value, the slopes are 0.040, 0.0432, 0.0483, 0.0497, and 0.0529. When the subjective weight and objective weight ratio is 0.3:0.7, the ideal point-posting progress value of each unit is relatively scattered. The resolution level is elevated, which is convenient for decision-making. However, the degree of optimization is not high, to facilitate the calculation. The combination weight is still calculated based on the equal contribution of the subjective weight and objective weight.

## Conclusions

1. The weighting choice has a considerable impact on the assessment of check dam threats in small watersheds. Three first-level indicators and ten second-level indicators that have an impact on risk assessment are determined using three features of flood catastrophe, operation, and economic risk. The analytical hierarchy process (AHP) and the entropy weight method are used to determine the subjective and objective weights of risk assessment indicators, and their combined weights are used to ensure the objectivity and impartiality of the evaluation indicators when computing the weights.

2. For the check dam system in the constrained watershed of the Loess Plateau, a combination weight TOPSIS-based risk assessment model has been developed. By applying the developed model to the check dam system units in Wangmaogou basin in China, the risk ranking result from the lowest to the highest risk is as follows: Wangmaogou 2# unit < Huangbaigou unit < Kanghegou unit < Nianyangou unit < Guandigou unit < Wangtagou unit. The risk ranking results are consistent with the actual situation.

3. The correlation between the ranking results generated by the combination weight method, AHP method, and entropy weight technique combined with TOPSIS was tested using the Spearman rank correlation coefficient test method. According to the correlation coefficient, which is not less than 0.943, the combination weight TOPSIS model efficiently integrates subjective experience assessment and objective data information. The idea is understandable, and the outcome makes sense.

4. In the given case, the outcomes of the combined weight TOPSIS model and the gray correlation theory's risk evaluation are consistent, although their linear slopes are 0.0483 and 0.0270, respectively. Therefore, in terms of evaluation resolution level and choice judgment, the combined weight TOPSIS model is more advantageous.

The factors affecting the safety of the check dam are not only the 10 indices selected in this paper, but also the selection of evaluation indices for the check dam connected by different operation states and different dam systems. Therefore, the selection of evaluation indices and the grade standards of indices should be further studied.

## Supporting information

**S1 Table.**
(CSV)

**S1 File.**
(XLSX)

**S2 File.**
(RAR)

## Author Contributions

**Conceptualization:** Lin Wang.

**Data curation:** Lin Wang.

**Formal analysis:** Lin Wang.

**Funding acquisition:** Lin Wang.

**Investigation:** Lin Wang.

**Methodology:** Lin Wang.

**Project administration:** Lin Wang.

**Resources:** Lin Wang.

**Software:** Lin Wang, Qiang Zu.

**Supervision:** Lin Wang.

**Validation:** Lin Wang, Qiang Zu.

**Visualization:** Lin Wang.

**Writing – original draft:** Lin Wang, Qiang Zu, Qiang Zhang.

**Writing – review & editing:** Lin Wang, Qiang Zu, Qiang Zhang.

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
