## [Decision Letter · Decision Letter 0]

4 Sep 2022

PONE-D-22-15836A risk assessment of check dam systems in small watersheds on the Loess Plateau: Taking the Wangmaogou watershed as an examplePLOS ONE

Dear Dr. lin,

Thank you for submitting your manuscript to PLOS ONE. After careful consideration, we feel that it has merit but does not fully meet PLOS ONE’s publication criteria as it currently stands. Therefore, we invite you to submit a revised version of the manuscript that addresses the points raised during the review process.

We look forward to receiving your revised manuscript.

Kind regards,

Pramod K Pandey

Academic Editor

PLOS ONE

Journal Requirements:

“The research work is supported by the National Natural Sciences Foundation of China Fund Project (Grant No.51909214), National Key R&D Program Project (2017YFC1501103) and Shaanxi Water Conservancy Science and Technology Plan Project(2021slkj-9).”

“No potential conflict of interest is reported by the authors.”

“The research work is supported by the National Natural Sciences Foundation of China Fund Project (Grant No.51909214), National Key R&D Program Project (2017YFC1501103) and Shaanxi Water Conservancy Science and Technology Plan Project(2021slkj-9).”

“The research work is supported by the National Natural Sciences Foundation of China Fund Project (Grant No.51909214), National Key R&D Program Project (2017YFC1501103) and Shaanxi Water Conservancy Science and Technology Plan Project(2021slkj-9).”

6. We note that Figures 1, 2 and 8 in your submission contain copyrighted images. All PLOS content is published under the Creative Commons Attribution License (CC BY 4.0), which means that the manuscript, images, and Supporting Information files will be freely available online, and any third party is permitted to access, download, copy, distribute, and use these materials in any way, even commercially, with proper attribution. For more information, see our copyright guidelines: http://journals.plos.org/plosone/s/licenses-and-copyright.

     1. You may seek permission from the original copyright holder of Figures 1, 2 and 8 to publish the content specifically under the CC BY 4.0 license.

7. We note that Figure 5 in your submission contain [map/satellite] images which may be copyrighted. All PLOS content is published under the Creative Commons Attribution License (CC BY 4.0), which means that the manuscript, images, and Supporting Information files will be freely available online, and any third party is permitted to access, download, copy, distribute, and use these materials in any way, even commercially, with proper attribution. For these reasons, we cannot publish previously copyrighted maps or satellite images created using proprietary data, such as Google software (Google Maps, Street View, and Earth). For more information, see our copyright guidelines: http://journals.plos.org/plosone/s/licenses-and-copyright.

    1. You may seek permission from the original copyright holder of Figure 5 to publish the content specifically under the CC BY 4.0 license.  

Additional Editor Comments:

Manuscript has some important information, however, it requires major revision. Please visit reviewer comments, and respond accordingly to be considered further.

Reviewers' comments:

Reviewer's Responses to Questions

**Comments to the Author**

1. Is the manuscript technically sound, and do the data support the conclusions?

Reviewer #1: Yes

Reviewer #2: Yes

2. Has the statistical analysis been performed appropriately and rigorously? 

Reviewer #1: No

Reviewer #2: Yes

3. Have the authors made all data underlying the findings in their manuscript fully available?

Reviewer #1: No

Reviewer #2: Yes

4. Is the manuscript presented in an intelligible fashion and written in standard English?

Reviewer #1: No

Reviewer #2: Yes

5. Review Comments to the Author

Reviewer #1: Comments:

The manuscript describes an approach to assess risk for check dam systems. The topic is interesting. However, I have a few suggestions :

Major suggestions:

• The title should be revised to “A Procedure for Risk Assessment of Check Dam Systems: A Case Study of Wangmaogou Watershed”

• The manuscript suggests a risk evaluation index system for check dam system using 10 indicators. However its applicability/reliability is not clear for the case or on what basis the author has chosen only 10 indicators (refer to Section 2.1)

• Similarly, section 2.2 assign a value for each C1…..C10 indicator. Are these values computed by a method or from any standard? The author should specify the sources.

• The author should add to the literature review an exciting paper on Entropy, AHP, and the combined weight-TOPSIS model for different applications.

• The authors use many different notations, some of which are not correctly defined or explained.

• Provide limitations (if any) of the study before the conclusion section.

• The conclusion section has redundant content that looks like a summary of the study. Remove and make it concise.

• I would like to recommend English editing in the manuscript strongly. There are a lot of errors in grammar and punctuation

Reviewer #2: Check dams are applied worldwide as an effective approach for soil conservation, However, failure risk of check dams restricts its development. The paper proposes a weighting method that combines the analytic hierarchy process and entropy method with TOPSIS to assess the risk of check dam systems. Although the research method of the paper is not innovative enough, the process and application of the method are introduced clearly, the content is rich, and it is valuable. Therefore, I recommend accepting this manuscript for publication, after some concerns are clarified with major revisions. My specific comments and suggestions are followed:

1. Manuscripts should be labeled with line numbers to help reviewers point out specific problems.

2. The research purpose of the paper should be introduced more clearly, and the conclusion should be more concise.

3. Why choose check dam system unit for evaluation instead of check dam system or single check dam?

4. Supplement the dividing standard of check dam system unit.

5. The evaluation results of check dam system units should be verified by actual investigation.

6. PLOS authors have the option to publish the peer review history of their article (what does this mean?). If published, this will include your full peer review and any attached files.

Reviewer #1: No

Reviewer #2: No

---

## [Author Response · Author response to Decision Letter 0]

19 Nov 2022

Ref.: Manuscript Number: PONE-D-22-15836

Title: A risk assessment of check dam systems in small watersheds on the Loess Plateau: Taking the Wangmaogou watershed as an example

Thanks for your positive comments. We have made minor changes based on Editor ’s comments.

Editor Comments:

Reply:

We have revised the manuscript according to the PLOS ONE style templates.

“The research work is supported by the National Natural Sciences Foundation of China Fund Project (Grant No.51909214), National Key R&D Program Project (2017YFC1501103) and Shaanxi Water Conservancy Science and Technology Plan Project(2021slkj-9).”

Reply:

The three funders participated in this study. The research work is supported by the National Natural Sciences Foundation of China Fund Project (Grant No.51909214), National Key R&D Program Project (2017YFC1501103) provided us with all the funds for the site investigation of the check dam. Shaanxi Water Conservancy Science and Technology Plan Project(2021slkj-9) participated in the data collection and analysis of this study.

“No potential conflict of interest is reported by the authors.”

Reply:

The authors have declared that no competing interests exist.

Reply:

Thank you very much for your reminder. I have uploaded the minimal underlying data set (i.e., Table4 Summary table of evaluation index, which used in the manuscript) to the GitHub in .csv format, and its URL is: 

https://github.com/XAUT-WangLin/Summary-table-of-evaluation-index/blob/main/Summary%20table%20of%20evaluation%20index.CSV. 

The uploaded data includes 7 evaluation indexes of 6 check dam systems involved in the manuscript. The 6 dam system includes Guandigou Unit、Wangtagou Unit、Wangmaogou Unit 2、Nianyangou Unit、Kanghegou Unit and Huangbaigou Unit. The 7 evaluation indexes are Detention flood、Remaining storage capacity、Drain facility、Dam integrity、Daily management、Emergency measures and Monitor facility.

All the calculations and conclusions in the study are completed through the above data and are reproducible.

“The research work is supported by the National Natural Sciences Foundation of China Fund Project (Grant No.51909214), National Key R&D Program Project (2017YFC1501103) and Shaanxi Water Conservancy Science and Technology Plan Project(2021slkj-9).”

“The research work is supported by the National Natural Sciences Foundation of China Fund Project (Grant No.51909214), National Key R&D Program Project (2017YFC1501103) and Shaanxi Water Conservancy Science and Technology Plan Project(2021slkj-9).”

Reply:

Sorry that funding information appears in the Acknowledgements section. We have deleted all funding information from the manuscript. 

6. We note that Figures 1, 2 and 8 in your submission contain copyrighted images. All PLOS content is published under the Creative Commons Attribution License (CC BY 4.0), which means that the manuscript, images, and Supporting Information files will be freely available online, and any third party is permitted to access, download, copy, distribute, and use these materials in any way, even commercially, with proper attribution. For more information, see our copyright guidelines: http://journals.plos.org/plosone/s/licenses-and-copyright.

 1. You may seek permission from the original copyright holder of Figures 1, 2 and 8 to publish the content specifically under the CC BY 4.0 license.

Reply:

Thank you very much for your reminder. We have replaced Figures 1, 2 and 8. These pictures were taken during our field research. We ensure that the replaced pictures are owned or created by us, and we have the original copyright. The new image is shown below. 

Fig 1. Check dams with ecological benefits（We own the copyright of this image）

Fig 2. Breach of check dams in Suide County（We own the copyright of this image）

Fig 8. Wangmaogou No. 2 backbone dam（We own the copyright of this image）

7. We note that Figure 5 in your submission contain [map/satellite] images which may be copyrighted. All PLOS content is published under the Creative Commons Attribution License (CC BY 4.0), which means that the manuscript, images, and Supporting Information files will be freely available online, and any third party is permitted to access, download, copy, distribute, and use these materials in any way, even commercially, with proper attribution. For these reasons, we cannot publish previously copyrighted maps or satellite images created using proprietary data, such as Google software (Google Maps, Street View, and Earth). For more information, see our copyright guidelines: http://journals.plos.org/plosone/s/licenses-and-copyright.

 1. You may seek permission from the original copyright holder of Figure 5 to publish the content specifically under the CC BY 4.0 license. 

Reply:

Thank you very much for your reminder. The base map in Figure 5 is download from the U.S. Geological Survey (USGS) - (http://www.usgs.gov). We created the base map, and we own the original copyright of the created image.

Ref.: Manuscript Number: PONE-D-22-15836

Title: A risk assessment of check dam systems in small watersheds on the Loess Plateau: Taking the Wangmaogou watershed as an example

Thanks for your positive comments. We have made minor changes based on reviewer 2’s comments.

Reviewer #1: Comments:

The manuscript describes an approach to assess risk for check dam systems. The topic is interesting. However, I have a few suggestions:

Major suggestions:

1.The title should be revised to “A Procedure for Risk Assessment of Check Dam Systems: A Case Study of Wangmaogou Watershed”.

Reply:

Corrected in Title. The title has been revised to read: “A Procedure for Risk Assessment of Check Dam Systems: A Case Study of Wangmaogou Watershed”, in P1 Line 1.

2.The manuscript suggests a risk evaluation index system for check dam system using 10 indicators. However, its applicability/reliability is not clear for the case or on what basis the author has chosen only 10 indicators (refer to Section 2.1).

Reply:

On P13 Line172 and 192, 3 references have been cited, and the indicators selected from the 3 references have been referred. Through comprehensive analysis, 10 indicators in the paper are selected as the safety evaluation indicators of check dam; At the same time, 3 references have been added to P33 Line562-566. Details of the literature are as follows：

42.Yang R, Li ZL, Wang D, Yuan SL, et al. Safety evaluation of check dam system in small watershed of loess plateau. Journal of Yan'an University (Natural Science Edition). 2018;37(01):41-45.(in Chinese)

43.Yuan SL, Zhang WH, Wang Z. Study on safety evaluation system of check dam system in small watershed in loess hilly region. Western Development (land development project research).2018;3(05):40-45.

44.Wang D, Ha YL, Li ZB, Yu KX, Bu CD, Zhang HW, Su LP, et al. Operational risk assessment of check dam system in ningxia typical watershed. Science of Soil and Water Conservation in China. 2017;15(03):17-25.(in Chinese)

3.Similarly, section 2.2 assign a value for each C1…..C10 indicator. Are these values computed by a method or from any standard? The author should specify the sources.

Reply: 

On P15 Line198, we calculated and assigned the values of each index according to the reference method cited, and added the reference cited on P33 Line569. Details of the literature are as follows：

45.Wang D, Li ZB, Li P, Gao HD, Zhao BH, Yuan SL, et al. Evaluation on layout of warping dam system in Jiuyuangou watershed. Research on water and soil conservation. 2016;23(05):49-55.(in Chinese)

4.The author should add to the literature review an exciting paper on Entropy, AHP, and the combined weight-TOPSIS model for different applications.

Reply:

In P5 Line101, a paper entitled " An Improved Entropy-Weighted Topsis Method for Decision-Level Fusion Evaluation System of Multi-Source Data " was cited and published in Entropy, and added the reference cited on P32 Line539. Details of the literature are as follows：

38. Liu L, Wan X, Li J, et al. An Improved Entropy-Weighted Topsis Method for Decision-Level Fusion Evaluation System of Multi-Source Data[J]. Sensors, 2022, 22(17): 6391.(in Chinese)

5.The authors use many different notations, some of which are not correctly defined or explained.

Reply:

In P8 line118, supplement n as the definition of the order of the judgment matrix Aij of the analytic hierarchy process.

In P9 line126, add the judgment matrix R that defines the entropy weight method.

In P9 line132, add the definition of entropy Sj of evaluation index.

In P10 line144, add X initial matrix.

In P11 line146, add the definition of positive exponent Pij.

In P11, add the definition of positive exponent Pij to line147.

The normalized matrix V is defined on P11 line151.

6.Provide limitations (if any) of the study before the conclusion section.

Reply:

Thank you very much for your comments. We are sorry that we did not clearly describe the limitations of the study. We have made the following modifications to the conclusion in P28 Line 402 Detail are as follows: "The factors affecting the safety of the check dam are not only the 10 indexes selected in this paper, but also the selection of evaluation indexes for the check dam connected by different operation states and different dam systems. Therefore, the selection of evaluation indexes and the grade standards of indexes should be further studied.”

7.The conclusion section has redundant content that looks like a summary of the study. Remove and make it concise.

Reply:

Corrected in conclusion to make it more concise. 

8.I would like to recommend English editing in the manuscript strongly. There are a lot of errors in grammar and punctuation.

Reply:

I’m sorry for that and the grammar and punctuation in the manuscript have been comprehensively, carefully and carefully revised. At the same time, we also invited native English speakers to polish the articles to meet the publishing requirements.

Reviewer #2: Comments:

Check dams are applied worldwide as an effective approach for soil conservation. However, failure risk of check dams restricts its development. The paper proposes a weighting method that combines the analytic hierarchy process and entropy method with TOPSIS to assess the risk of check dam systems. Although the research method of the paper is not innovative enough, the process and application of the method are introduced clearly, the content is rich, and it is valuable. Therefore, I recommend accepting this manuscript for publication, after some concerns are clarified with major revisions. My specific comments and suggestions are followed:

1.Manuscripts should be labeled with line numbers to help reviewers point out specific problems.

Reply:

Line numbers have been marked on the manuscript according to your comments. Thank you very much for your valuable comments.

2.The research purpose of the paper should be introduced more clearly, and the conclusion should be more concise.

Reply:

As you said, the research purpose of the paper is not clear enough. First of all, we modify Line 55-59 in Introduction as“However, the check dam system is an indispensable method to control soil erosion and prevent floods on the Loess Plateau [19-22]. If the check dam breaks, it will cause huge loss of life and property to the local people [23]. Comprehensive risk assessment can be carried out to realize the risk ranking of the check dam system in small basins, which can reduce the risk caused by the burst of the check dam to the greatest extent [24].”

The conclusions have been simplified, and the present paper conclusions are as follows:

（1）The rating of check dams' danger in small watersheds is significantly influenced by the weighting decision. Three aspects of flood catastrophe, operation, and economic risk are used to determine three first-level indicators and ten second-level indicators that affect risk assessment. The subjective and objective weights of risk assessment indicators are obtained in accordance with the analytical hierarchy process (AHP) and the entropy weight method, and their combined weights are obtained to guarantee the objectivity and impartiality of the evaluation indicators when calculating the weights.

(2) The combination weight TOPSIS-based risk assessment model for the check dam system in the Loess Plateau's limited watershed is established. In the little watershed of Wangmaogou, the model is used. The Wangtagou unit poses the highest risk in the event of a once-in-a-century rainfall, and the results of the risk ranking are essentially accurate.

(3) The correlation between the ranking results generated by the combination weight method, AHP method, and entropy weight technique combined with TOPSIS is tested using the Spearman rank correlation coefficient test method. The combination weight TOPSIS model effectively integrates subjective experience assessment and objective data information, according to the correlation coefficient, which is not less than 0.943. The idea is understandable, and the outcome makes sense.

(4) In the given case, the outcomes of the combined weight TOPSIS model and the grey correlation theory's risk evaluation are consistent, although their linear slopes are 0.0483 and 0.0270, respectively. Therefore, in terms of evaluation resolution level and choice judgment, the combined weight TOPSIS model is more advantageous.

3.Why choose check dam system unit for evaluation instead of check dam system or single check dam?

Reply:

The check dam system is complex and densely distributed, and the check dam unit can be used to quickly and easily divide the complex dam system into some simple and easy to evaluate units, such as References 40 and 44. Of course, for the warping dam in a large basin, the basin can be divided into multiple warping dam systems for risk assessment. For more accurate management and safety considerations, a single check dam can also be selected for risk assessment, such as References 42 and 43.

4.Supplement the dividing standard of check dam system unit.

Reply:

The check dam is a water and soil conservation engineering measure built on the gully. In the check dam system, it is believed that there is a main ditch and several branch ditches, and a large backbone dam will be built on the main ditch. Taking the branch ditch connected to the main ditch as a unit, the warping dam on the branch ditch is divided into a warping dam unit. The warping dam on the branch ditch will be named by the name of the branch ditch in turn, as shown in Figure 5 (d) of P17 Line238: a branch ditch in Wangmao Valley is Nianyan ditch, and the warping dams (4, 5, 6) on the branch ditch will be named Nianyan ditch 1 dam, Nianyan ditch 2 dams, Nianyan ditch 3 dams, which will be divided into a unit (Nianyan ditch unit).

5.The evaluation results of check dam system units should be verified by actual investigation.

Reply:

Thank you very much for your valuable advice. Lines306-322 of the paper are the results of our teams' actual investigation: In 2012, when the rainstorm hit Wangmaogou watershed on July 15, a total of 8 check dams broke. In Wangtagou unit dam, the Wangtagou dam 1# and dam 2# broke. While WangMaogou Dam 2 did not break.

The following statement is added after P22 Line 323:“According to the actual survey after the “7.15” rainstorm. Neither of the two check dams in Unit 2 of Wangmaogou break during the rainstorm. In the Huangbaigou Unit, the Huangbaigou 2# dam did not break, and the Huangbaigou 2# dam break is circular with breach of 2m. There are three check dams in the Nianyangou Unit, two of them have broken. There are also three check dams in Kanghegou Unit, and all of them have broken. Among them, only the 3# dam of Kanghegou has broken up to 7m wide, and the breaking is relatively serious. The two check dams in Wangtagou Unit broke seriously, and the maximum breach reached 11.2m wide.” Through the actual investigation after the rainstorm, the evaluation result " Wangmaogou Unit 2<Huangbaigou Unit<Nianyangou Unit<Kang Hegou unit<Guandigou unit<Wangtagou unit" of the siltation dam unit is verified to be correct.

Added references to actual investigations in P33 Line 571: [49] Zhang Z, Chai J, Li Z, et al. Reconstruction and effects of a failure of a typical check dam system under an extreme rainstorm on the Loess Plateau, China[J]. Natural Hazards, 2022, 111(2): 1401-1419. ”

---

## [Editor Report · Decision Letter 1]

28 Nov 2022

PONE-D-22-15836R1A Procedure for Risk Assessment of Check Dam Systems: A Case Study of Wangmaogou WatershedPLOS ONE

Dear Dr. lin,

Thank you for submitting your manuscript to PLOS ONE. After careful consideration, we feel that it has merit but does not fully meet PLOS ONE’s publication criteria as it currently stands. Therefore, we invite you to submit a revised version of the manuscript that addresses the points raised during the review process.

We look forward to receiving your revised manuscript.

Kind regards,

Pramod K Pandey

Academic Editor

PLOS ONE

Additional Editor Comments (if provided):

This manuscript has some novelty, however, major revision is needed. Authors should work on improving following points.

1. Avoid unnecessary equations.

2. Write clearly: goal and objectives

3. In method section, author should write clearly how/which methods are used in achieving those goals.

4. Research and discussion should be focused on findings of this study and supported by external references.

5. Conclusions and abstract should be improved.

6. Improvement in English of manuscript.
---

## [Author Response · Author response to Decision Letter 1]

16 Feb 2023

Ref.: Manuscript Number: PONE-D-22-15836R2

Title: A Procedure for Risk Assessment of Check Dam Systems: A Case Study of Wangmaogou Watershed

Thanks for your positive comments. We have responded to Editor ’s question.

Editor ’s Questions:

1. Thank you for your response. Please update the Figure captions of Figure 5 to include attribution to ESRI. Once this is done, we should be able to proceed.

Reply:

Thank you for your comment. The Figure caption of Figure 5 in Line 246, Page16 now has been modified to “Fig 5. Location of the area (a, b); the Wangmaogou watershed (c, Drawn by ESRI's ArcGIS); the layout of check dam types (d).”

---

## [Decision Letter · Decision Letter 2]

19 Apr 2023

PONE-D-22-15836R2A Procedure for Risk Assessment of Check Dam Systems: A Case Study of Wangmaogou WatershedPLOS ONE

Dear Dr. lin,

Thank you for submitting your manuscript to PLOS ONE. After careful consideration, we feel that it has merit but does not fully meet PLOS ONE’s publication criteria as it currently stands. Therefore, we invite you to submit a revised version of the manuscript that addresses the points raised during the review process.

We look forward to receiving your revised manuscript.

Kind regards,

Pramod K Pandey

Academic Editor

PLOS ONE

Reviewers' comments:

Reviewer's Responses to Questions

**Comments to the Author**

1. If the authors have adequately addressed your comments raised in a previous round of review and you feel that this manuscript is now acceptable for publication, you may indicate that here to bypass the “Comments to the Author” section, enter your conflict of interest statement in the “Confidential to Editor” section, and submit your "Accept" recommendation.

Reviewer #2: All comments have been addressed

Reviewer #3: All comments have been addressed

2. Is the manuscript technically sound, and do the data support the conclusions?

Reviewer #2: Yes

Reviewer #3: Yes

3. Has the statistical analysis been performed appropriately and rigorously? 

Reviewer #2: Yes

Reviewer #3: Yes

4. Have the authors made all data underlying the findings in their manuscript fully available?

Reviewer #2: No

Reviewer #3: Yes

5. Is the manuscript presented in an intelligible fashion and written in standard English?

Reviewer #2: Yes

Reviewer #3: Yes

6. Review Comments to the Author

Reviewer #2: (No Response)

Reviewer #3: Author should correct a few identified typographical and grammatical errors noticed in the manuscript. For instance on line 38, a word lagre is supposed to be large.

7. PLOS authors have the option to publish the peer review history of their article (what does this mean?). If published, this will include your full peer review and any attached files.

Reviewer #2: No

Reviewer #3: No

---

## [Author Response · Author response to Decision Letter 2]

8 May 2023

Ref.: Manuscript Number: PONE-D-22-15836R2

Title: A Procedure for Risk Assessment of Check Dam Systems: A Case Study of Wangmaogou Watershed

Comments to the Author 

4. Have the authors made all data underlying the findings in their manuscript fully available?

The PLOS Data policy requires authors to make all data underlying the findings described in their manuscript fully available without restriction, with rare exception (please refer to the Data Availability Statement in the manuscript PDF file). The data should be provided as part of the manuscript or its supporting information, or deposited to a public repository. For example, in addition to summary statistics, the data points behind means, medians and variance measures should be available. If there are restrictions on publicly sharing data—e.g., participant privacy or use of data from a third party—those must be specified.

Reviewer #2: No

Reviewer #3: Yes

Reply:

Thank you for your comment. 

All the data has been uploaded to https://github.com/XAUT-WangLin/Summary-table-of-evaluation-index. It includes 1 copy of the Wangmaogou DEM file, 1 copy of Summary table of evaluation index of Wangmaogou watershed, and 1 copy of calculating table. These data completely include the topographic data used in the paper, the raw data used in the calculation of the paper, and the data of the whole process of the thesis operation. The data is open source and can be downloaded and used via the connection above.

6. Review Comments to the Author

Reviewer #2: (No Response)

Reviewer #3: Author should correct a few identified typographical and grammatical errors noticed in the manuscript. For instance, on line 38, a word lagre is supposed to be large.

Reply:

Thanks for your comments. We have revised manuscript carefully and the details are as follws:

In Line 38, “lagre” has been corrected to “large”.

In Line 320, “The” has been corrected to “the”.

In Line 390, “Huangbogou” has been corrected to “Huangbaigou”.

All the “Kang Hegou” has been corrected to “Kanghegou”.

---

## [Editor Report · Decision Letter 3]

31 May 2023

PONE-D-22-15836R3A Procedure for Risk Assessment of Check Dam Systems: A Case Study of Wangmaogou WatershedPLOS ONE

Dear Dr. lin,

Thank you for submitting your manuscript to PLOS ONE. After careful consideration, we feel that it has merit but does not fully meet PLOS ONE’s publication criteria as it currently stands. Therefore, we invite you to submit a revised version of the manuscript that addresses the points raised during the review process.

We look forward to receiving your revised manuscript.

Kind regards,

Pramod K Pandey

Academic Editor

PLOS ONE

Journal Requirements:

Additional Editor Comments (if provided):

This manuscript is acceptable, however, substantial editing in terms of formatting, and improvement in sentence structuring need to be improved. Also make sure that paragraphs are well aligned. Further, please remove the figure caption's writing that author owns the copy right. If author burrowed then they should write the source otherwise no need to write copy right. Also, most figure caption requires little bit more information to elaborate figure caption. Thank you for revision.
---

## [Author Response · Author response to Decision Letter 3]

4 Jun 2023

Ref.: Manuscript Number: PONE-D-22-15836R3

Title: A Procedure for Risk Assessment of Check Dam Systems: A Case Study of Wangmaogou Watershed

Additional Editor Comments:

This manuscript is acceptable, however, substantial editing in terms of formatting, and improvement in sentence structuring need to be improved. Also make sure that paragraphs are well aligned. Further, please remove the figure caption's writing that author owns the copy right. If author burrowed then they should write the source otherwise no need to write copy right. Also, most figure caption requires little bit more information to elaborate figure caption. Thank you for revision.

reply:

Thanks for your positive advice and we are so glad to receive the message that you would like to accept this manuscript. We also revise our manuscript according to your letter, and the details are as follow:

1. The figure caption's writing that author owns the copy right now has been removed from Fig.1, Fig.2 and Fig.8.

2. The Fig.5 caption now has been revised as: “Fig 5. Location of the area (a, b); tthe altitude of the Wangmaogou watershed (c, Drawn by ESRI's ArcGIS); the layout of check dam types (d).”

3. Some words and the figure size has been adjusted to make Table.1 and Table.3 more suitable for pages.

---

## [Editor Report · Decision Letter 4]

13 Jun 2023

A Procedure for Risk Assessment of Check Dam Systems: A Case Study of Wangmaogou Watershed

PONE-D-22-15836R4

Dear Dr. lin,

We’re pleased to inform you that your manuscript has been judged scientifically suitable for publication and will be formally accepted for publication once it meets all outstanding technical requirements.

Kind regards,

Pramod K Pandey

Academic Editor

PLOS ONE

Additional Editor Comments (optional):

Manuscript is acceptable in current form for publication.
---

## [Editor Report · Acceptance letter]

19 Jun 2023

PONE-D-22-15836R4 

A Procedure for Risk Assessment of Check Dam Systems: A Case Study of Wangmaogou Watershed 

Dear Dr. Wang:

I'm pleased to inform you that your manuscript has been deemed suitable for publication in PLOS ONE. Congratulations! Your manuscript is now with our production department. 

Kind regards, 

on behalf of

Dr. Pramod K Pandey 

Academic Editor

PLOS ONE